# PersonaEval: Are LLM Evaluators Human Enough to Judge Role-Play?

**Lingfeng Zhou**[1], **Jialing Zhang**[1], **Jin Gao**[1], **Mohan Jiang**[1,2], **Dequan Wang**[1,2]*
[1]Shanghai Jiao Tong University    [2]Shanghai Innovation Institute

## Abstract

Current role-play studies often rely on unvalidated LLM-as-a-judge paradigms, which may fail to reflect how humans perceive role fidelity. A key prerequisite for human-aligned evaluation is role identification, the ability to recognize who is speaking based on dialogue context. We argue that any meaningful judgment of role-playing quality (how well a character is played) fundamentally depends on first correctly attributing words and actions to the correct persona (who is speaking). We present PersonaEval, the first benchmark designed to test whether LLM evaluators can reliably identify human roles. PersonaEval uses human-authored dialogues from novels, scripts, and video transcripts, challenging models to determine the correct persona according to the conversation context. Our experiments, including a human study, show that even the best-performing LLMs reach only around 69% accuracy, well below the level needed for reliable evaluation. In contrast, human participants perform near ceiling with 90.8% accuracy, highlighting that current LLM evaluators are still not human enough to effectively judge role-play scenarios. To better understand this gap, we examine training-time adaptation and test-time compute, suggesting that reliable evaluation requires more than task-specific tuning, but depends on strong, human-like reasoning abilities in LLM evaluators. We release our benchmark at https://github.com/maple-zhou/PersonaEval.

## 1 Introduction

A growing body of work in role-play adopts LLM-as-a-judge paradigms, where models are tasked with evaluating the role-playing behavior of other models (Shao et al., 2023; Wang et al., 2024c; 2025b; Lu et al., 2024). While scalable, this strategy assumes that large language models (LLMs) can approximate human judgment, a claim that remains largely untested. This gap in validation raises concerns about the reliability and true human-alignment of current evaluation pipelines. Recent studies have already revealed misalignment between LLMs and human, including preference leakage (Ghasemi et al., 2025; Murugadoss et al., 2024), where models favor outputs from their own model family. What's more, Zhao et al. (2025) show that one token is enough to fool LLM judges. In a broader cognitive sense, Josh Tenenbaum argues that LLMs derive intelligence from language, while humans develop language after acquiring intelligence (Cherian et al., 2024). These gaps call into question whether current LLMs can reliably assess role fidelity in a human-like way.

A key prerequisite for human-aligned evaluation is the ability to identify the speaker's role from context. We argue this is a foundational capability: an LLM cannot credibly assess how well a role is portrayed if it cannot first determine who is speaking. This is because accurate identification is essential for grounding the interpretation of dialogue and avoiding critical errors, such as misattributing behaviors or inconsistencies to the wrong speaker. Such failures ultimately compromise the fairness and precision of the entire evaluation process, making role identification a minimal, yet objective, test of alignment with human.

To examine this, we introduce **PersonaEval**, the first benchmark for evaluating whether LLMs can reliably identify character roles from dialogue context. The task is formulated

---

*Corresponding author: dequanwang@sjtu.edu.cn

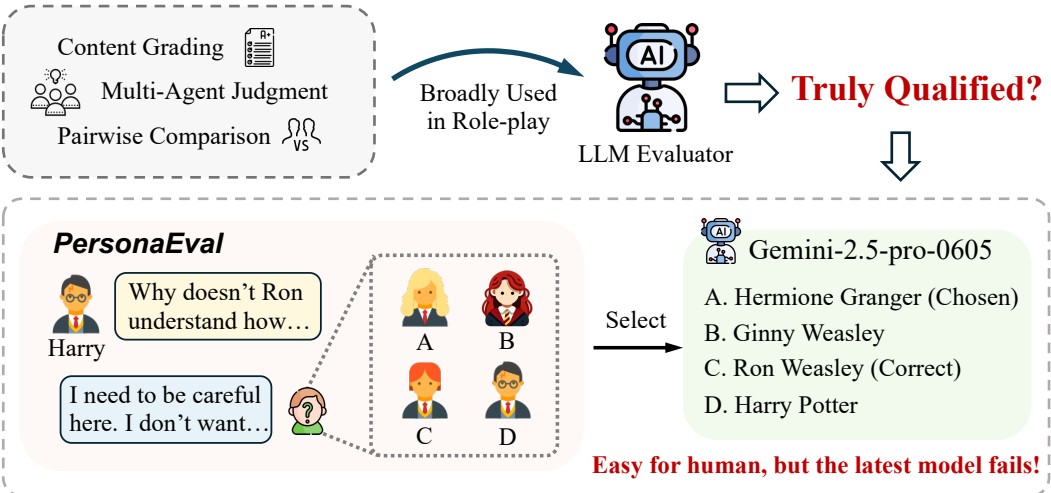

Figure 1: LLM-as-a-judge paradigms are broadly used in role-play evaluation, while its reliability has not been verified. We propose PersonaEval, the first benchmark to investigate a necessary condition for human-aligned LLM evaluators of role-play: accurate identification of roles from dialogue context. We find that even the latest model Gemini-2.5-pro-0605[2] fails the case which is easy for humans (Section 4.3), indicating the challenge remains unsolved.

as a constrained classification: given a dialogue snippet and four candidate roles, the evaluator must select the role most consistent with the target utterance (Figure 1). We ground our evaluation in human judgment by curating instances from human-authored materials—including novels, scripts, and transcripts—and supplement them with detailed role descriptions to mitigate potential LLM knowledge gaps.

Our experiments reveal a clear limitation of LLMs. State-of-the-art LLMs achieve no more than 69% accuracy on PersonaEval, which falls short of meeting a convincing baseline for reliable assessment, and even the latest model fails on trivial cases (Figure 1). In contrast, human participants achieve 90.8% accuracy. These findings offer a clear answer to the question in our title: LLM evaluators are not yet "human enough" to judge role-play.

To understand what makes a better human-aligned LLM evaluator, we investigate two common strategies: training-time adaptation and test-time compute. Surprisingly, we find fine-tuning LLMs with role-play data does not improve performance and can even degrade it, suggesting that memorizing role-specific knowledge is insufficient. In contrast, test-time methods show more potential. Notably, reasoning models consistently outperform others.

These results suggest that strong role-play evaluation depends less on simple heuristics and more on robust, context-aware reasoning. Accurately judging roles requires inference, perspective-taking, and social understanding—skills more aligned with human judgment than with pattern matching. Taken together, our findings point to test-time compute, particularly inference-time reasoning, as a promising direction for building LLM evaluators of role-play that better reflect human-like judgment. Our main contributions are as follows:

- We introduce PersonaEval, the first benchmark to directly evaluate whether LLMs can identify human roles from natural dialogue, a necessary but underexplored foundation for reliable role-play evaluation.

- We demonstrate that state-of-the-art LLMs fall short of human-level performance on role identification, revealing a critical gap in their ability to reflect human judgment.

- We find that reliable role-play evaluation depends not on role-specific training or prompting, but on reasoning ability, highlighting test-time compute especially reasoning as a promising strategy for building more human-aligned LLM evaluators.

---

[2]This experiment was performed in June 2025, when Gemini-2.5-pro-0605 was the latest model.

## 2 Related Work

**Role-playing Evaluation**   Role-playing is essential for aligning large language models (LLMs) with human values, which positions human experts as the gold standard for evaluation, just like what early LLM role-play studies do (Zhou et al., 2023). However, the vast scale of modern role-playing systems makes thorough human evaluation impractical due to high costs and delays. Therefore, some works (Lu et al., 2025; Wang et al., 2024a;b) turn to traditional metrics of natural language generation tasks, including BLEU and ROUGE. These metrics compute similarities between the reference text and the generated content. Yet in the domain of role-play, ground truth is always uncertain, since there are multiple possible correct responses. As a result, researchers gradually shift to reference-free evaluation. Some works train a reward model to estimate the quality of generated responses (Tu et al., 2024; Wu et al., 2025; Dai et al., 2024). More and more works are using LLMs as judges to evaluate the quality of role-play content from a human perspective (Gusev, 2024; Wang et al., 2024c), benefiting from their strong generalization ability and capacity to provide more comprehensive and fine-grained feedback. While some of them leverage LLM evaluators in a reference-guided way (Zhou et al., 2024a; Yu et al., 2024), most utilize LLMs to score from diverse dimensions (Yang et al., 2025; Lu et al., 2025; Zhou et al., 2024a; Shao et al., 2023) or execute pairwise comparisons (AI, 2024; Wu et al., 2025). Among them,  Wang et al. (2025b) even builds a multi-agent system to make the final judgment. To avoid the subjectivity bias originated from scoring, some works let LLM evaluators perform multi-class classification (Lu et al., 2024; Yuan et al., 2024).

These approaches aim to assess how effectively LLMs play specific roles, contributing to a broader understanding of their performance in role-playing scenarios. However, challenges remain in ensuring that evaluation results are both reliable and aligned with human expectations of role adherence.

**Evaluating LLM Evaluator**   The reliability of LLMs as evaluators is gaining increasing attention in the community (Son et al., 2024a; Wei et al., 2024; Zhang et al., 2024). Researchers start by testing the instruction-following abilities of LLMs in their role as evaluators.  Zheng et al. (2023) addresses this issue by using advanced LLMs to assess performance on open-ended questions.  Wang et al. (2023) approach ChatGPT as a human-like evaluator by providing task-specific (e.g., summarization) and aspect-specific (e.g., relevance) instructions to prompt ChatGPT for evaluating the outputs of various NLG (Natural Language Generation) models.  Murugadoss et al. (2024) explores whether LLM assessments are based purely on prompt instructions or also reflect inherent preferences for high-quality data similar to their fine-tuning data. There are also various meta-evaluation studies of LLMs with different focuses.  Chern et al. (2024) evaluates LLM evaluators through multi-agent debates, while  Son et al. (2024b) and  Hada et al. (2024) propose multilingual benchmarks.  Eiras et al. (2025) carefully examines the safety-related aspects of LLM judges.  Zhou et al. (2024b) pays attention to the domain of paper review, while  Wang et al. (2025a) cares about the performance of LLM evaluators in software engineering.

Although interest in the meta-evaluation of LLMs is growing, the specific evaluation of LLMs as role-play evaluators remains underexplored. While some work (Yang et al., 2025) includes ablation studies to examine alignment between LLM evaluators and human experts, there is still a lack of a systematic perspective on meta-evaluating role-play LLM evaluators, particularly regarding their ability to distinguish between nuanced role identities. Yet this ability is fundamental to higher-order role-play evaluation; without it, any subsequent judgment of role fidelity rests on an unstable and unverified foundation, revealing a critical gap in our understanding of how well LLMs can perform human-aligned assessment.

## 3 PersonaEval

We design PersonaEval to test whether large language models (LLMs) can perform human-like judgment in a core subtask of role-play evaluation: identifying who a speaker is based on dialogue context. To build a reliable benchmark for role identification, we start by formulating the task as a constrained classification problem (Section 3.1), then construct diverse,

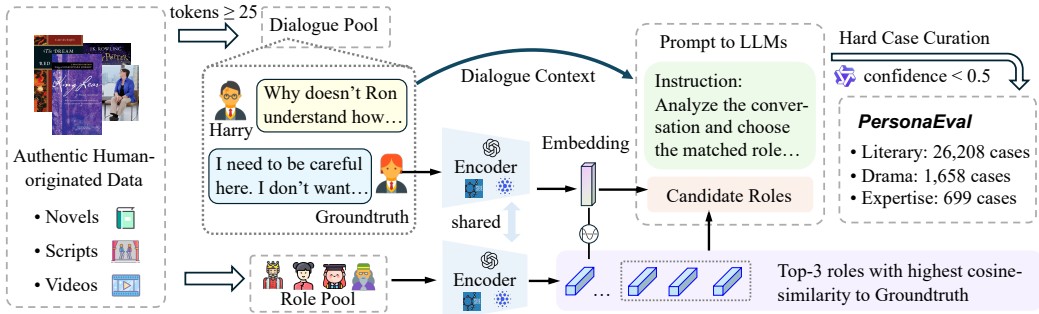

Figure 2: Curation pipeline of PersonaEval. PersonaEval evaluates role identification ability of LLM evaluators via a constrained classification task built on human-authored dialogues from novels, scripts, and videos. Each instance includes a two-turn conversation: the first turn is from a known character, and the second is from a target character whose identity is to be predicted by LLMs. The ground truth role's embedding is used to retrieve the top-3 most similar roles from a candidate pool, forming a challenging candidate set. A standardized prompt (See Appendix A) presents the dialogue and candidate roles to LLMs. To ensure difficulty, only instances where a strong baseline model (Qwen-max) shows low confidence in the correct answer are retained. The final benchmark includes three tracks—PersonaEval-Literary (26,208), PersonaEval-Drama (1,658), and PersonaEval-Expertise (699).

human-authored datasets that capture role-play across different domains (Section 3.2). To increase challenge and minimize shortcut cues, we design semantically similar adversarial distractors (Section 3.3). Finally, we apply a hard case curation pipeline (Section 3.4). See Figure 2 for the complete curation pipeline.

## 3.1 Task Formulation

PersonaEval adopts a classification framework to evaluate role identification capabilities in LLMs. Each instance presents a two-turn dialogue between two characters: Character1 (with a known identity) and Character2 (whose identity is to be inferred). Models are given the dialogue context and four candidate roles (five for expertise-level settings, see Section 3.3), each accompanied by a detailed profile. Evaluators assign confidence scores to each candidate, with higher scores indicating stronger belief in a match. To mitigate positional bias, candidate order is randomized. This design turns an inherently subjective task into a verifiable one, using deterministic ground truth derived from source materials. The full prompt is provided in Appendix A.

While this setting may appear less common, this is largely because prior work has often implicitly assumed that LLMs possess such basic capabilities. Our intention is to carefully examine this assumption and bridge a gap that, to our knowledge, has been under-explored. Nonetheless, normally in a role-play conversation, the context we have is more than just 2 turns. We agree that extending evaluations to richer, multi-turn contexts is important, and we view our work as a necessary first step toward that broader goal.

## 3.2 Data Composition

PersonaEval includes three tracks, each targeting different facets of role understanding:

- **PersonaEval-Literary**: Built on 26,208 dialogues from 771 English novels, this track tests persona inference in fictional narratives. The data are curated from CoSER (Wang et al., 2025b), a verified fiction-based dataset containing texts from classic and modern novels.

- **PersonaEval-Drama**: This track contains 1,658 Chinese dialogue snippets from screenplays, testing models' ability to understand role alignment in scripted in-

teractions. The data are adapted from the partially open-source CharacterEval datasets (Tu et al., 2024).

- **PersonaEval-Expertise**: Sourced from the Wired "5 Levels" video series[3], this track features 699 scaffolded explanations where domain experts tailor content to audiences of different knowledge levels (Child, Teen, College Student, Graduate Student, and Expert). Dialogues test whether models can infer a speaker's intended audience based on linguistic and conceptual cues, without relying on specialized domain knowledge.

Together, these tracks ensure coverage across fictional, performative, and instructional domains. Literary texts emphasize character personality and inner thoughts, while scripts focus more on conversational style, and instructional videos involve role-audience alignment. This diversity helps mitigate domain-specific bias to some extent.

Importantly, all source data are human-authored, avoiding contamination from synthetic model-generated content. This design choice ensures evaluation aligns with human, as public datasets increasingly overlap with prior model outputs.

We also balance language diversity with the availability and quality of human-authored materials, providing two English tracks and one Chinese track. English resources, especially literary novels, are more abundant and varied, allowing us to construct robust, high-quality benchmarks. At the same time, we include a Chinese track to introduce linguistic and cultural diversity, recognizing the importance of evaluating models in non-English contexts.

Since some materials may not be included in the pretraining corpus of LLMs, we provide detailed role descriptions, including both basic traits and nuanced plot-related context. This helps ensure that role identification performance reflects reasoning ability rather than missing background knowledge, minimizing the impact of knowledge gaps.

### 3.3 Adversarial Distractor Construction

To ensure each instance presents a genuine reasoning challenge, we construct adversarial distractors that are semantically close to the correct role. As shown in Figure 2, for PersonaEval-Literary and PersonaEval-Drama, we embed all candidate role profiles using three independently trained models to avoid bias from single model structure: OpenAI's text-embedding-3-small (OpenAI, 2024), BGE-M3 (Chen et al., 2024), and Sentence-BERT's (Reimers & Gurevych, 2019) multilingual distiluse-base-multilingual-cased-v1[4]. For each instance, we compute cosine similarity between the ground-truth role profile and all others in the pool. From each embedding model, we select the top non-target role (i.e., most similar but incorrect) and use these to form a distractor set of three, ensuring diverse but challenging contrastive options. This forces evaluators to resolve subtle, human-level ambiguities, rather than rely on surface heuristics.

In the PersonaEval-Expertise track, we leverage the fixed five-tier role hierarchy (Child to Expert). Each dialogue uses the four incorrect levels as distractors, reflecting authentic educational scaffolding challenges without needing embedding-based selection. All options are paired with audience-specific profiles drawn from the original video transcripts.

Across all tracks, we avoid synthetic perturbations and instead build distractors from naturally occurring human roles. This ensures that the task tests real-world ambiguity resolution, not artificial contrastive tricks.

### 3.4 Hard Case Curation

We observe that the original unfiltered data contain numerous trivial cases (e.g., direct name mentions, simple greetings) that do not genuinely test role inference. To avoid inflating performance with trivial cases and ensure the benchmark focuses on meaningful reasoning, we explicitly filter for difficult instances where even strong models struggle. Our goal is not

---

[3]https://www.wired.com/video/series/5-levels
[4]https://huggingface.co/sentence-transformers/distiluse-base-multilingual-cased-v1

broad coverage, but a focused evaluation of whether models can resolve subtle, human-level ambiguities. We apply a two-stage filtering process:

**Stage 1: Low-Information Filtering** We remove dialogue turns where Character2's utterance is under 25 tokens. These low-information responses (e.g., "Exactly.") offer little basis for role inference and are excluded to maintain meaningful task complexity.

**Stage 2: Confidence-Based Filtering** Using Qwen-max (Bai et al., 2023), we filter out instances where the model shows high confidence in the correct answer. Specifically, we discard any instance where Qwen-max assigns over 50% confidence to the ground-truth role. From an initial pool of over 110,000 instances, this filtering yields 28,565 challenging examples, each requiring inference beyond surface-level cues. Nonetheless, this confidence-based filtering may introduce systematic bias. More discussion can be found in Appendix B.

## 4 Experiment

We organize our experimental analysis into four parts. We first evaluate model performance on PersonaEval (Section 4.1), then analyze the impact of reasoning ability of LLMs (Section 4.2), present case studies (Section 4.3), and conclude with a human study (Section 4.4).

Since prior work often uses competent large language models (LLMs) like GPT-3.5 and GPT-4 as role-play evaluators (Shao et al., 2023; Lu et al., 2024), we evaluate a diverse set of state-of-the-art models, including several reasoning ones rarely considered in existing role-play studies. Model versions are specified in Appendix C. We report classification accuracy (top-1 accuracy) as the primary metric for model performance.

### 4.1 Main Results on Role Identification with PersonaEval

Figure 3 summarizes model performance across the three tracks of PersonaEval. Most LLMs achieve around 40-60% accuracy, including the latest GLM-4.5, and even the best-performing model, Gemini-2.5-pro (03-25 version, as detailed in Appendix C), reaches only 68.8%, well below human-level performance (see Section 4.4). These findings reveal a significant gap between current LLMs and the requirement of consistent role identification. As this task is a necessary condition for credible role-play evaluation, the results highlight a core limitation of LLM-as-a-judge methods. Full results are provided in Appendix D.

Nonetheless, some models show signs of progress. As shown in Table 1 placed in Appendix D, top-2 accuracy and mean rank suggest partial understanding: many models rank the correct role second, suggesting they are still aware of the correct answer. Notably,

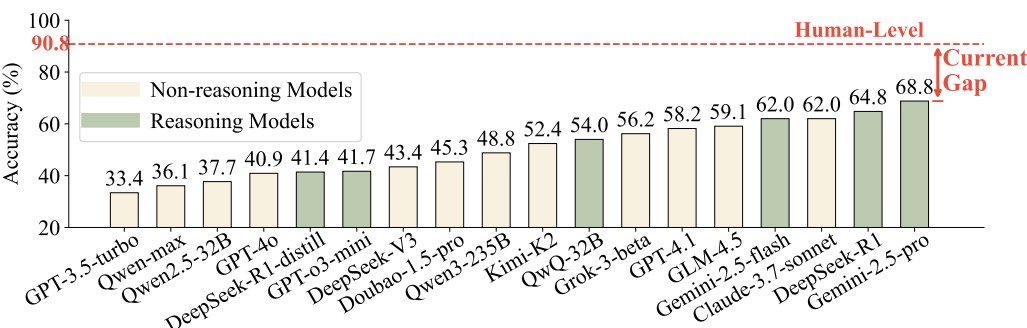

Figure 3: Accuracy of LLMs on PersonaEval. Most LLMs struggle with role identification compared to human, including the latest GLM-4.5, highlighting a fundamental limitation. Reasoning models show clear advantages, suggesting that effective evaluation requires deeper reasoning. DeepSeek-R1-distill refers to DeepSeek-R1-distill-Qwen-32B.

Claude-3.7-sonnet, DeepSeek-R1, and Gemini-2.5-pro achieve near-perfect top-2 accuracy. Furthermore, both ECE and Brier Score remain relatively low, indicating that models express appropriate uncertainty rather than overconfident errors. Together, these results provide a more optimistic view, suggesting that while LLM evaluators are not yet human-level, some are catching up. With further advances, they may become viable for role-play evaluation.

## 4.2 Analysis on Reasoning Models

Reasoning models show clear advantages over base models in role identification. As shown in Figure 3, Gemini-2.5-pro outperforms all foundation models, including Claude-3.7-sonnet.

We also compare reasoning models with and without native reasoning capabilities. DeepSeek-R1-distill-Qwen-32B, which distills reasoning from DeepSeek-R1 into Qwen2.5, achieves only a small gain over its base model. In contrast, QwQ-32B, trained via reinforcement learning on reasoning-intensive tasks, significantly outperforms Qwen2.5. This suggests that shallow or distilled reasoning may not transfer well to role-play evaluation, while end-to-end reasoning training is more effective.

These results support a broader insight: effective role identification requires robust, model-native reasoning. Beyond pattern matching, evaluators must engage in contextual judgment and logical inference—capabilities more aligned with human-like evaluation.

Our analysis suggests that several specific types of reasoning are particularly important for effective role identification. These include:

- Perspective-taking: The ability to infer the speaker's background, goals, and point of view based on contextual clues within the dialogue.

- Intent inference: The capacity to understand the underlying intention behind an utterance, which often goes beyond its surface-level semantic meaning.

- Pragmatic reasoning: The skill of interpreting the social and contextual meanings of statements as they unfold within an interaction.

These cognitive skills allow an evaluator to move past simple linguistic pattern matching and instead build a coherent model of the speaker's identity, which is fundamental for judging role-play fidelity in a human-aligned manner.

## 4.3 Case Study

Through case studies, we find that many examples, which are trivial for humans, are frequently misinterpreted by LLMs. In these cases, models often follow the wrong trajectory of reasoning. We suspect this is due to a fundamental difference in how role-play is interpreted: LLMs tend to focus on surface-level linguistic cues such as speaking style, while humans perform deeper reasoning. Specifically, humans excel at intent inference, allowing them to prioritize the speaker's communicative goals, and employ pragmatic reasoning to understand the social context of the dialogue, abilities that current models still lack. This observation echoes Josh Tenenbaum's view that LLMs derive intelligence from language, whereas humans develop language after acquiring intent-driven cognition.

We illustrate one such case below. The bracketed text denotes the character's internal thoughts. For a human reader, the reasoning path to identify the speaker is direct. First, Harry's internal monologue explicitly names "Ron" as the subject of his concern ("Why doesn't Ron understand..."). Second, the respondent (Character2) directly addresses "Harry" in his reply. By integrating these two explicit references, the reader can unambiguously conclude that Character2 is Ron Weasley. However, the advanced model fails to connect these critical contextual signals, instead appearing to focus on the speaker's tone and linguistic style. Additional cases are provided in Appendix E.

| Dialogue and Candidate Roles | Response from **Gemini-2.5-pro-0605** |
|---|---|
| **Character1: Harry Potter**
[Thinking: **Why doesn't Ron understand** how important this is?]
I want to know what he's up to. And don't tell me it's all in my head, not after what I overheard between him and Snape.
**Character2: Unknown**
[Thinking: I need to be careful here. **I don't want to dismiss Harry's concerns**, but this is getting out of hand.]
I never said it was all in your head, but there's no rule saying only one person at a time can be plotting anything in this place! **You're getting a bit obsessed with Malfoy, Harry**.

**Candidate Roles:**
Hermione Granger, Ginny Weasley, **Ron Weasley (Correct)**, and Harry Potter | **Analysis:**
**Hermione Granger is the most likely responder**, as her cautious internal thought, logical reasoning ("no rule saying..."), and balanced confrontation ("You're getting a bit obsessed...") align closely with her character profile.
Ginny is a plausible second due to her confidence and perceptiveness. **Ron is less likely given the measured tone** that doesn't quite match his typically more reactive style. Harry is Character 1 being the one expressing obsession and therefore cannot be the responder.

**Final Answer:**
**Hermione: 0.75**, Ginny: 0.15, **Ron: 0.10**, Harry: 0.00
(Higher probability indicates stronger role correspondence.) |

## 4.4 Human Study

To evaluate human performance on PersonaEval and compare it with LLMs, we conduct a controlled study with 20 highly educated volunteers, including 10 undergraduates and 10 PhD students, selected to match the task's reasoning and domain knowledge demands. Each participant is given 50 examples sampled from cases where DeepSeek-R1 makes incorrect high-confidence predictions, spanning all three tracks of the benchmark. No time limits are imposed to allow for careful, thoughtful responses.

We deliberately select challenging cases for human evaluation to ensure meaningful comparisons on non-trivial instances. While this sampling introduces distributional differences, it is actually a conservative choice that substantially strengthens our findings: if humans still perform well on cases that are consistently difficult for models, the performance gap becomes even more compelling. We have manually verified that many cases in the benchmark are straightforward for humans. If we had sampled uniformly across the full benchmark, human accuracy would likely have been even higher.

As shown in Figure 3, participants achieve an average accuracy of 90.8%, far surpassing the best-performing LLM. This highlights a substantial gap between current LLM capabilities and human-level understanding in role identification. Full results are in Appendix F.

## 5 Improvement Investigation

To better understand what contributes to effective role-play evaluation, we investigate two common strategies for improving LLMs: training-time adaptation and test-time compute. At training time, we test whether injecting role-specific knowledge through fine-tuning improves role identification performance (Section 5.1). At test time, we apply few-shot prompting and self-consistency to examine whether additional inference-time computation can enhance evaluator accuracy (Section 5.2). Detailed results can be found in Appendix G.

## 5.1 Training-time Adaptation

Many role-play studies fine-tune LLMs to inject role-specific knowledge, aiming to improve downstream performance. In this section, we evaluate two generations of such models:

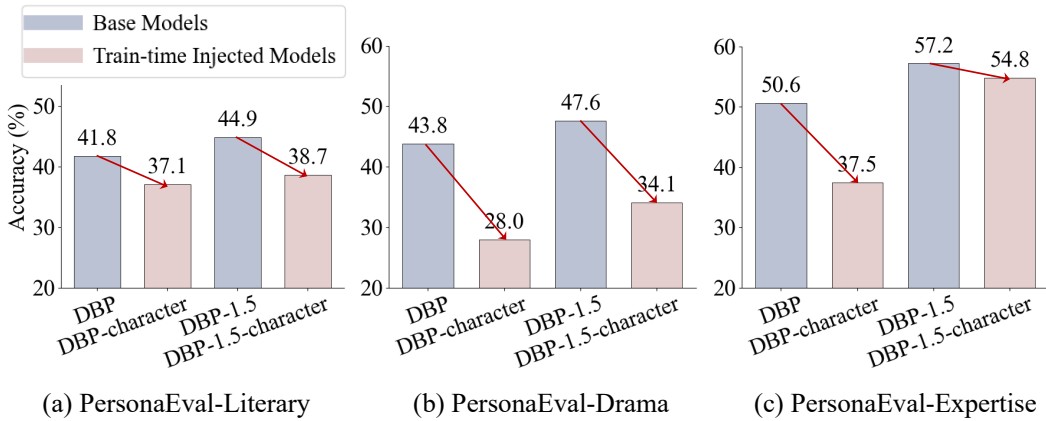

(a) PersonaEval-Literary     (b) PersonaEval-Drama     (c) PersonaEval-Expertise

Figure 4: Accuracy of 4 models on the three tracks of PersonaEval, corresponding to (a), (b), and (c), suggests that fine-tuning on role-specific data does not improve role identification performance and may even degrade it, while improvements in base model capability show more consistent gains. DBP is short for Doubao-pro. DBP-character and DBP-1.5-character are fine-tuned from DBP and DBP-1.5 respectively using role-specific data.

Doubao-pro-character and Doubao-1.5-pro-character, which are obtained by fine-tuning Doubao-pro and Doubao-1.5-pro, respectively.

These models are closed-source and provided by ByteDance[5]. The training-time adaptation is performed by fine-tuning on datasets enriched with role knowledge, identity recognition, dialogue alignment, behavior alignment, and instruction tuning. These are implemented under a broader System Prompt + SFT framework to ensure that models can maintain persona consistency, recognize social roles, and infer identity throughout dialogue interactions. In addition, the system also emphasizes multi-turn dialogue memory, context retention, and user portrait construction to support long-term coherence.

As shown in Figure 4, fine-tuning on role knowledge does not improve performance on our role identification task; in fact, it degrades it. A possible explanation is that memorizing role-related patterns during fine-tuning may interfere with the model's native reasoning ability, which is essential for evaluating nuanced persona alignment. This suggests that injecting role knowledge through training-time adaptation is not an effective way to build reliable role-play evaluators. In contrast, comparing Doubao-pro and Doubao-1.5-pro shows that improvements in the base model itself, such as general capability upgrades, have a more positive impact on role identification performance.

We also test CoSER-Llama3.1-8B (Wang et al., 2025b), a fine-tuned model introduced in the data source of PersonaEval-Literary. However, it fails to follow our benchmark instructions and cannot even produce usable outputs, making evaluation infeasible. This further highlights that role-specific training alone is insufficient for constructing capable evaluators.

## 5.2 Test-time Compute

In this section, we evaluate two popular test-time compute strategies: few-shot prompting (Brown et al., 2020) and self-consistency (Wang et al., 2022). We focus on PersonaEval-Literary using Qwen-max and DeepSeek-V3, as this track forms the core of PersonaEval and offers more stable shot selection than others.

For few-shot prompting, we test 1-shot, 3-shot, and 5-shot settings. For self-consistency, we apply majority voting with $K = 3$ and $K = 5$, as well as a weighted average. As shown in Figure 5, few-shot prompting consistently improves model performance, but gains plateau by 5-shot. In contrast, self-consistency provides negligible improvement across all settings.

---

[5]https://team.doubao.com/en

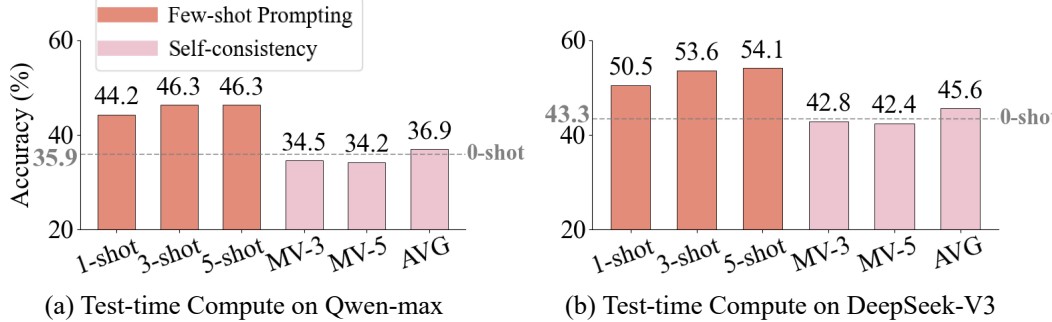

(a) Test-time Compute on Qwen-max          (b) Test-time Compute on DeepSeek-V3

Figure 5: Accuracy on PersonaEval-Literary using two test-time compute strategies on (a) Qwen-max and (b) DeepSeek-V3, respectively. The gray dashed line in each plot indicates the 0-shot accuracy of the corresponding model. Few-shot prompting (1-shot, 3-shot, 5-shot) shows consistent but saturated gains. In contrast, self-consistency (MV: majority vote, AVG: weighted average) yields negligible improvement across settings.

These findings support our earlier analysis (Section 4.2): role-play evaluation relies more on a model's reasoning ability than on sampling or ensembling techniques. While few-shot prompting can impart some reasoning patterns, its effect is constrained by the quality and generalizability of exemplars. Self-consistency, meanwhile, merely reinforces a model's existing reasoning without extending it, yielding no meaningful performance boost.

## 6   Conclusion

In this work, we introduce PersonaEval, the first benchmark for evaluating whether LLMs can reliably identify character roles from natural dialogue, a foundational step toward human-aligned role-play evaluation. Our results show that current LLM evaluators perform well below human-level, with even the strongest and latest models struggling on cases that are trivial for humans. This highlights a core limitation in existing LLM-as-a-judge pipelines. Through empirical analysis, we further find that training-time adaptation with role-specific data offers little benefit, while test-time methods, especially reasoning models, show more promise. These findings suggest that robust, context-aware reasoning, rather than memorization, is essential for effective role-play evaluation, and point to test-time compute as a practical and forward-looking direction.

Looking ahead, a promising line of future work is to move beyond output accuracy and investigate how LLM evaluators arrive at their predictions. This includes analyzing reasoning trajectories across models, comparing them with human thought processes, and identifying where their judgments diverge. To support this, new diagnostic tools may be developed to visualize and interpret model reasoning paths. Such insights can inform new forms of test-time guidance, such as human-aligned reasoning chains or rationale-based prompting. Another direction is to explore how human-like reasoning strategies can be systematically injected into models at inference time. Ultimately, closing the gap between LLM and human evaluation may require bridging deeper cognitive differences—moving beyond pattern recognition toward reasoning grounded in intent, context, and social understanding, much like how human judgment emerges from experience before language, as emphasized by Josh Tenenbaum. We hope PersonaEval serves as a step toward this goal.

## Acknowledgments

This research is supported by the Key R&D Program of Shandong Province, China (2023CXGC010214, 2024CXGC010213). We express our gratitude to the funding agency for their support. We thank all the anonymous reviewers for their valuable suggestions.

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

# A   Prompt to LLM Evaluators

---

### An Example Prompt for Evaluating LLMs in PersonaEval

You are an AI specialist tasked with dialogue role recognition. Please analyze the following conversation and determine the likelihood of four character candidates being the responder.
**# Conversation**
[Character1: Harry Potter]
[Begin]
[Why doesn't Ron understand how important this is?] (frustrated) I want to know what he's up to. And don't tell me it's all in my head, not after what I overheard between him and Snape.
[End]
[Character2]
[Begin]
[I need to be careful here. I don't want to dismiss Harry's concerns, but this is getting out of hand.] I never said it was all in your head, but there's no rule saying only one person at a time can be plotting anything in this place! You're getting a bit obsessed with Malfoy, Harry.
[End]
**# Task**
Your task is to analyze the response of Character2 and estimate the Bayesian probability distribution for each of the four character candidates. The probabilities should reflect the likelihood of each candidate being the one responding, based on their profiles. The sum of all probabilities must equal 1. A higher probability for a candidate indicates that the response aligns better with that candidate.
**# Character Candidates**
**1. Ginny Weasley**
Ginny Weasley, the youngest child and only daughter of the Weasley family, emerges as a pivotal character in "Harry Potter and the Half-Blood Prince." With her fiery red hair and strong-willed personality, Ginny has grown from Ron's shy little sister into a confident and capable young witch. She possesses a quick wit and a mischievous streak, often using humor to diffuse tense situations...
**2. Ron Weasley**
Ron Weasley, Harry Potter's loyal best friend and steadfast companion, plays a crucial role in the final installment of the Harry Potter series. With his trademark red hair and freckles, Ron embodies the heart and humor of the trio. Coming from a large, loving wizarding family, Ron's background provides both strength and insecurity as he faces the challenges ahead...
**3. Hermione Granger**
Hermione Granger, in "Harry Potter and the Deathly Hallows", emerges as a brilliant, resourceful, and fiercely loyal young witch. As one of Harry Potter's closest friends and allies, she plays a crucial role in the quest to defeat Lord Voldemort. Hermione's exceptional intelligence and vast magical knowledge make her an invaluable asset to the trio's mission...
**4. Harry Potter**
Harry Potter, the protagonist of "Harry Potter and the Half-Blood Prince," is a sixteen-year-old wizard entering his sixth year at Hogwarts School of Witchcraft and Wizardry. With his distinctive lightning bolt scar and round glasses, Harry continues to bear the weight of being "The Chosen One" in the fight against Lord Voldemort...
**# Response Format**
Analyze step by step, and then output the following JSON object containing the final probability distribution. Ensure that the sum of all probabilities equals 1, with each probability representing the likelihood that a given candidate is the responder. Do not modify the character names, and use the full character names exactly as they appear in the task.

```json
{
    "Ginny Weasley": probability_for_Ginny Weasley,
    "Ron Weasley": probability_for_Ron Weasley,
    "Hermione Granger": probability_for_Hermione Granger,
    "Harry Potter": probability_for_Harry Potter,
}
```

---

## B    Discussion of Confidence-Based Filering

Our choice of an automated, single-model filtering strategy is motivated by the large scale of the dataset, which makes manual annotation or more complex multi-model validation schemes impractical within the scope of this work. The unfiltered data include numerous trivial cases (e.g., direct name mentions, simple greetings) that do not genuinely test role inference. Thus, a curation process is essential to ensure the benchmark's focus on reasoning.

We recognize the potential for this method to introduce systematic bias. The resulting dataset is most reliably considered a collection of hard cases for models of the Qwen family or those with similar capabilities. Importantly, we do not directly select cases where Qwen fails, but rather use a confidence threshold (less than 0.5 for the correct answer) to identify uncertain cases, reducing overfitting the filter to any particular model. However, our primary objective in this paper is to uncover and demonstrate an overlooked reasoning problem in LLMs. The curated dataset, by successfully challenging even a powerful model like Qwen-max, serves this purpose effectively and provides strong support for our main claims. We believe this benchmark, despite its limitations, is a valuable first step. For future applications aiming for a more generalized benchmark, we recommend adopting a cross-validation approach using a diverse suite of models to further enhance robustness and utility.

## C    Versions of LLMs Used in the Paper

All LLMs with its repective version is as follows:

- GPT-3.5-turbo (OpenAI, 2023): gpt-3.5-turbo-0125
- GPT-4o (OpenAI, 2023): gpt-4o-2024-08-06
- GPT-4.1 (OpenAI, 2025a): gpt-4.1-2025-04-14
- GPT-o3-mini (OpenAI, 2025b): o3-mini-2025-01-31
- Qwen-max (Bai et al., 2023): qwen-max-2025-01-25
- Qwen3-235B (Team, 2025a): Qwen3-235B-a22B, non-thinking version
- Doubao-1.5-pro[6]: doubao-1-5-pro-32k-250115
- DeepSeek-V3 (Bi et al., 2024): DeepSeek-V3-250324
- DeepSeek-R1 (Guo et al., 2025): DeepSeek-R1-0120
- Gemini-2.5-pro (Team et al., 2024): Gemini-2.5-pro-preview-03-25

Other models, including Claude-3.7-sonnet (Anthropic, 2025), Qwen2.5-32B (Yang et al., 2024), QwQ-32B (Team, 2025b), DeepSeek-R1-distill-Qwen-32b (Guo et al., 2025), Kimi-K2-Instruct (Team et al., 2025), GLM-4.5 (Z.AI, 2025), Grok-3-beta (xAI, 2025), and Gemini-2.5-flash (Team et al., 2024), so far there's only one version.

## D    Detailed Experiment Results of PersonaEval

We report the detailed experiment results of PersonaEval here, including top-1 accuracy, top-2 accuracy, mean rank (MR) of the ground truth role in responses, Expected Calibration Error (ECE), and Brier Score (BS). Except mean rank, others are shown in percentage. DeepSeek-R1-distill refers to DeepSeek-R1-distill-Qwen-32B. The best performance in each column is marked in bold.

We display the metrics summarized on the whole benchmark first in Table 1, and then the results on the three track are shown respectively. We also compare the two versions of DeepSeek-V3, indicating that the capability gap of role-play LLM evaluators are narrowing.

Performance across models appears strongly correlated with the model capability. For instance, GPT-3.5-turbo consistently underperforms relative to more advanced models,

---

[6]https://team.doubao.com/en

while Claude-3.7-sonnet demonstrates significantly superior performance. These results align with expectations, reaffirming the importance of model quality in evaluator tasks.

Furthermore, model specialization plays a notable role. For example, GPT-4o demonstrates competitive performance in PersonaEval-Expertise, on par with Claude-3.7-sonnet, while DeepSeek-V3 performs particularly well on benchmarks targeting Chinese-language tasks. These observations suggest that role-play evaluator competence is not only model-dependent but also domain-specific.

| Model | Top-1 Acc ↑ | Top-2 Acc ↑ | MR ↓ | ECE ↓ | BS ↓ |
|---|---|---|---|---|---|
| GPT-3.5-turbo | 33.4 | 71.6 | 2.02 | 21.1 | 19.5 |
| Qwen-max | 36.1 | 77.7 | 1.92 | 28.7 | 20.8 |
| Qwen2.5-32B | 37.7 | 77.7 | 1.89 | 23.5 | 19.1 |
| GPT-4o | 40.9 | 83.2 | 1.75 | 16.0 | 16.7 |
| DeepSeek-R1-distill | 41.4 | 75.0 | 1.92 | 22.6 | 19.2 |
| GPT-o3-mini | 41.7 | 76.3 | 1.88 | 23.4 | 18.5 |
| DeepSeek-V3-241226 | 38.2 | 81.0 | 1.80 | 24.4 | 18.7 |
| DeepSeek-V3-250324 | 43.4 | 84.2 | 1.73 | 22.8 | 17.8 |
| Doubao-1.5-pro | 45.3 | 82.0 | 1.72 | 21.1 | 17.3 |
| Qwen3-235B | 48.8 | 84.8 | 1.70 | 20.3 | 16.6 |
| Kimi-K2 | 52.4 | 86.9 | 1.60 | 26.6 | 16.8 |
| QwQ-32B | 54.0 | 83.8 | 1.67 | 22.3 | 16.4 |
| Grok-3-beta | 56.2 | 91.0 | 1.50 | **4.4** | 12.7 |
| GPT-4.1 | 58.2 | 91.3 | 1.47 | 11.9 | 12.4 |
| GLM-4.5 | 59.1 | 87.6 | 1.54 | 9.7 | 13.7 |
| Gemini-2.5-flash | 62.0 | 89.9 | 1.46 | 11.9 | 12.6 |
| Claude-3.7-sonnet | 62.0 | 91.2 | 1.46 | 8.3 | 12.1 |
| DeepSeek-R1 | 64.8 | 90.0 | 1.48 | 14.9 | 13.2 |
| Gemini-2.5-pro | **68.8** | **92.6** | **1.38** | 4.9 | **10.5** |

Table 1: Complete results across all three tracks of PersonaEval.

| Model | Top-1 Acc ↑ | Top-2 Acc ↑ | MR ↓ | ECE ↓ | BS ↓ |
|---|---|---|---|---|---|
| GPT-3.5-turbo | 33.5 | 70.9 | 2.06 | 21.3 | 19.5 |
| Qwen-max | 35.9 | 76.3 | 1.98 | 29.2 | 21.0 |
| Qwen2.5-32B | 38.0 | 75.7 | 1.96 | 23.7 | 19.2 |
| GPT-4o | 41.3 | 83.7 | 1.79 | 16.0 | 16.6 |
| DeepSeek-R1-distill | 41.8 | 72.9 | 1.98 | 23.0 | 19.2 |
| GPT-o3-mini | 42.5 | 76.9 | 1.88 | 22.9 | 18.4 |
| DeepSeek-V3-241226 | 38.0 | 79.5 | 1.90 | 24.6 | 18.7 |
| DeepSeek-V3-250324 | 43.3 | 83.5 | 1.78 | 22.9 | 17.8 |
| Doubao-1.5-pro | 44.9 | 79.1 | 1.84 | 21.6 | 17.5 |
| Qwen3-235B | 49.5 | 86.0 | 1.67 | 20.0 | 16.5 |
| Kimi-K2 | 53.0 | 88.1 | 1.59 | 26.5 | 16.7 |
| QwQ-32B | 54.3 | 82.1 | 1.71 | 23.0 | 16.5 |
| Grok-3-Beta | 57.1 | 92.3 | 1.47 | **4.0** | 12.4 |
| GPT-4.1 | 59.6 | 93.1 | 1.44 | 11.1 | 12.1 |
| GLM-4.5 | 60.9 | 89.5 | 1.51 | 10.1 | 13.5 |
| Gemini-2.5-flash | 62.5 | 91.0 | 1.45 | 11.9 | 12.5 |
| Claude-3.7-sonnet | 63.0 | 91.8 | 1.48 | 8.0 | 11.8 |
| DeepSeek-R1 | 65.4 | 89.4 | 1.49 | 14.7 | 13.1 |
| Gemini-2.5-pro | **69.5** | **93.3** | **1.37** | 5.4 | **10.3** |

Table 2: Complete results on PersonaEval-Literary.

| Model | Top-1 Acc ↑ | Top-2 Acc ↑ | MR ↓ | ECE ↓ | BS ↓ |
|---|---|---|---|---|---|
| GPT-3.5-turbo | 26.3 | 57.8 | 2.33 | 26.1 | 21.8 |
| Qwen-max | 31.6 | 66.9 | 2.16 | 33.4 | 22.5 |
| Qwen2.5-32B | 30.0 | 61.9 | 2.24 | 29.6 | 21.6 |
| GPT-4o | 28.3 | 61.3 | 2.24 | 26.1 | 20.9 |
| DeepSeek-R1-distill | 30.1 | 61.5 | 2.24 | 26.8 | 21.3 |
| GPT-o3-mini | 35.0 | 63.9 | 2.14 | 30.7 | 21.0 |
| DeepSeek-V3-241226 | 34.7 | 63.5 | 2.19 | 30.3 | 21.8 |
| DeepSeek-V3-250324 | 42.2 | 71.4 | 1.98 | 30.0 | 20.2 |
| Doubao-1.5-pro | 47.6 | 75.3 | 1.85 | 22.6 | 17.4 |
| Qwen3-235B | 38.7 | 72.1 | 1.97 | 29.0 | 19.9 |
| Kimi-K2 | 47.8 | 78.8 | 1.78 | 33.3 | 19.7 |
| QwQ-32B | 48.0 | 76.6 | 1.82 | 19.3 | 16.8 |
| Grok-3-Beta | 46.9 | 78.6 | 1.80 | 16.3 | 16.6 |
| GPT-4.1 | 45.4 | 76.3 | 1.84 | 23.5 | 17.5 |
| GLM-4.5 | 47.2 | 78.5 | 1.80 | 17.9 | 16.6 |
| Gemini-2.5-flash | 56.7 | 81.1 | 1.67 | 20.5 | 15.6 |
| Claude-3.7-sonnet | 50.2 | 77.0 | 1.80 | 18.6 | 16.7 |
| DeepSeek-R1 | 56.3 | 84.1 | 1.64 | 22.4 | 15.8 |
| Gemini-2.5-pro | **60.3** | **85.8** | **1.57** | **13.1** | **13.4** |

Table 3: Complete results on PersonaEval-Drama.

| Model | Top-1 Acc ↑ | Top-2 Acc ↑ | MR ↓ | ECE ↓ | BS ↓ |
|---|---|---|---|---|---|
| GPT-3.5-turbo | 45.1 | 69.8 | 2.05 | 10.0 | 14.2 |
| Qwen-max | 51.5 | 77.3 | 1.85 | 6.0 | 12.6 |
| Qwen2.5-32B | 47.6 | 73.1 | 1.99 | 6.7 | 13.2 |
| GPT-4o | 56.7 | 78.5 | 1.79 | 9.1 | 12.0 |
| DeepSeek-R1-distill | 50.5 | 73.4 | 1.96 | 7.7 | 13.4 |
| GPT-o3-mini | 29.8 | 53.8 | 2.41 | 23.5 | 16.9 |
| DeepSeek-V3-241226 | 50.8 | 73.8 | 1.92 | 7.9 | 13.0 |
| DeepSeek-V3-250324 | 51.6 | 74.1 | 1.93 | 3.6 | 12.8 |
| Doubao-1.5-pro | 57.2 | 82.0 | 1.74 | 10.2 | 12.0 |
| Qwen3-235B | 47.2 | 67.7 | 2.08 | 10.4 | 14.1 |
| Kimi-K2 | 39.6 | 60.8 | 1.76 | 14.8 | 13.6 |
| QwQ-32B | 57.2 | 77.5 | 1.84 | **4.2** | 12.2 |
| Grok-3-beta | 43.2 | 69.2 | 2.04 | 8.5 | 13.5 |
| GPT-4.1 | 34.7 | 56.3 | 1.87 | 13.1 | 14.0 |
| GLM-4.5 | 44.4 | 62.3 | 1.69 | 8.9 | 12.5 |
| Gemini-2.5-flash | 55.3 | 69.8 | **1.40** | 12.7 | **10.2** |
| Claude-3.7-sonnet | 53.8 | 74.8 | 1.86 | 9.6 | 12.0 |
| DeepSeek-R1 | 60.9 | 80.7 | 1.70 | 5.9 | 10.9 |
| Gemini-2.5-pro | **62.1** | **84.4** | 1.62 | 12.1 | 11.0 |

Table 4: Complete results on PersonaEval-Expertise.

# E Thorough Case Study

We present several cases in which competent LLMs make high-confidence errors, while the task remains relatively straightforward for humans. The bracketed text denotes the character's internal thoughts.

## E.1 Case 1: Proper Addressing

This case is from *Harry Potter and the Goblet of Fire (Harry Potter, #4)*. It involves character addressing, which requires some thought from humans but is relatively straightforward. LLMs, however, are easily misled by the "Weasley" clue and perform extensive, unnecessary analysis in the wrong direction.

**Proper Analysis:** Character2 cannot be Hermione Granger (option 4), as she is already speaking as Character1. Severus Snape (option 3) would never refer to Mr. Weasley in such a casual or thoughtful way, nor would he show curiosity about dramatic student entrances. Ron Weasley (option 1) is also unlikely to be Character2, because he would not refer to his own father as "Mr. Weasley"—he would say "Dad". This detail strongly suggests that Character2 is Harry Potter (option 2), who often refers to Ron's father as "Mr. Weasley" and was present at the Quidditch World Cup to hear that quote firsthand. The tone—thoughtful and slightly amused—also fits Harry, who frequently reflects on past conversations and tries to make sense of unusual magical situations. Therefore, Harry is the most fitting choice for Character2.

| Dialogue and Candidate Roles | Response from Claude-3.7-sonnet |
|---|---|
| **Character1: Hermione Granger**
[Thinking: I'm getting a bit frustrated having to repeat this information.]
You can't Apparate inside the Hogwarts grounds, how often do I have to tell you?
**Character2: Unknown**
[Thinking: I remember what Mr. Weasley said about wizards showing off. Maybe that's what they'll do.]
Maybe the foreign students are preparing a dramatic entrance... Remember what Mr. Weasley said at the World Cup? 'Always the same — we can't resist showing off when we get together...'

**Candidate Roles:**
Ron Weasley, Harry Potter (Correct), Severus Snape, and Hermione Granger | **Analysis:**
The response most likely comes from Ron Weasley, who naturally quotes his father, attended the World Cup, and speaks in a casual, speculative tone. Harry Potter is a secondary possibility, having attended the World Cup, but lacks Ron's personal connection and typical commentary style. Severus Snape is highly unlikely, as the tone, content, and familiarity with Mr. Weasley don't match his character. Hermione Granger is explicitly Character1 and speaks differently, making her an impossible fit.

**Final Answer:**
Ron: 0.85, Harry: 0.15, Snape: 0.00, Hermione: 0.00 |

## E.2 Case 2: Character Traits

This case is from *The Adventures of Tom Sawyer*. It requires comparing key characteristics of the candidate characters and selecting the most appropriate one. For humans, as long as they are familiar with the rough traits of the options, making the correct choice is not difficult. However, LLMs often become overly focused on character profiles and plot details, leading to unnecessary complexity in their reasoning.

**Proper Analysis:** Character2 cannot be Aunt Polly (option 1), as she is an adult and wouldn't be involved in a secretive moral dilemma with Huck in this context—especially not discussing running away or saving Muff Potter. Sid (option 3), Tom's well-behaved and rather tattletale-like cousin, lacks the courage and internal conflict shown here; he's

unlikely to risk trouble to help someone like Muff Potter. Huckleberry Finn (option 2) is already speaking as Character1, so he cannot also be Character2. That leaves Tom Sawyer (option 4), who fits perfectly. The line reflects Tom's classic internal struggle between fear of consequences and his growing sense of justice—a major theme in The Adventures of Tom Sawyer. Tom is the one who knows Muff Potter is innocent and eventually risks his safety to reveal the truth, making him the clear match for Character2.

| Dialogue and Candidate Roles | Response from GPT-4o |
|---|---|
| **Character1: Huckleberry Finn** [Thinking: I try to think of a way out of this mess.] Maybe we ought to leave town, Tom. Just light out and never come back. **Character2: Unknown** [Thinking: I'm torn between fear and a sense of responsibility.] But what about Muff Potter? He didn't do nothing, and he's gonna hang for it if we don't say something. **Candidate Roles:** Aunt Polly, Huckleberry Finn, Sid, and Tom Sawyer (Correct) | **Analysis:** The response reflects fear and moral responsibility, best matching Huckleberry Finn, who witnessed the event and cares deeply about justice. Tom Sawyer also fits, given his eventual decision to defend Muff Potter. Aunt Polly is unlikely, since she stays removed from such events, and Sid lacks both the involvement and moral weight needed for this response. **Final Answer:** Polly: 0.05, Huckleberry: 0.60, Sid: 0.05, Sawyer: 0.30 |

### E.3 Case 3: Irrelevant Characters

This case is from *The Adventures of Huckleberry Finn*. In this case, identifying the irrelevant options based on the dialogue is sufficient to make the correct choice quickly, without needing to analyze expressive details. This makes the task relatively simple for humans. However, LLMs tend to focus on language style and plot, overlooking the simplest relationship between the characters in the dialogue.

**Proper Analysis:** Character2 cannot be Huck Finn (option 1) or Huckleberry Finn (option 3), as he is already speaking as Character1. The use of "we" and the contrast in their tones show that two distinct characters are involved. Uncle Tom (option 2) is not a character in *The Adventures of Huckleberry Finn*—he belongs to a completely different novel (*Uncle Tom's Cabin* by Harriet Beecher Stowe)—so he can be ruled out entirely. That leaves Tom Sawyer (option 4), which fits perfectly. Tom is known for his elaborate, imaginative plans, even when they're impractical or dangerous—like digging Jim out with a case-knife. In this passage, he's beginning to compromise slightly due to the urgency of the situation, but still clings to the idea of a "proper" escape. His language, full of roundabout logic and excitement about the adventure, is classic Tom Sawyer.

| Dialogue and Candidate Roles | Response from DeepSeek-V3 |
|---|---|
| **Character1: Huckleberry Finn** [Thinking: I can't believe we're really going to do this.] All right—I don't care where he comes out, so he comes out; and Jim don't, either, I reckon. But there's one thing, anyway—Jim's too old to be dug out with a case-knife. He won't last. **Character2: Unknown** [Thinking: We need to stick to the plan, but maybe we can compromise a little.] | **Analysis:** The response shows pragmatic and strategic thinking, which aligns closely with Huckleberry Finn's practical mindset. The informal dialect ("resk" instead of "risk") and collaborative tone further support this match. Tom Sawyer, by contrast, tends to propose elaborate, romanticized plans based on adventure novels, which doesn't match the straightforward, cautious tone of this response. |

Well, we can't resk being as long as we ought to, because it mayn't take very long for Uncle Silas to hear from down there by New Orleans. He'll hear Jim ain't from there. Then his next move will be to advertise Jim, or something like that. So we can't resk being as long digging him out as we ought to.

**Candidate Roles:**
Huck Finn, Uncle Tom, Huckleberry Finn, and Tom Sawyer (Correct)

Uncle Tom, from a completely different novel (Uncle Tom's Cabin), lacks any connection to the context, characters, or dialect here. Since "Huck Finn" and "Huckleberry Finn" are duplicate entries for the same character, the high likelihood is split between them unless they are merged.

**Final Answer:**
Huck: 0.50, Uncle: 0.00, Huckleberry: 0.50, Sawyer: 0.00

### E.4 Case 4: Tone and Complex Plot Inference

This case is from *A Storm of Swords (A Song of Ice and Fire, #3)*. It is more challenging, primarily due to the domain knowledge required. However, with a general understanding of the plot of the book, humans can make the correct choice through reasoning and analysis. For LLMs, this case requires identifying the right direction and careful reasoning. However, LLMs often follow incorrect reasoning paths and prematurely conclude an incorrect answer based solely on the "Kingslayer" clue.

**Proper Analysis:** Character2 cannot be Tywin Lannister (options 1 and 4), as Tywin is known for his stern, authoritative demeanor and would never speak with such sarcasm or self-deprecation—especially not about being a kinslayer or kingslayer. Jaime Lannister (option 2) is famously the "kingslayer" (mentioned in the profile), but he has never been to Castle Black, which makes the reference to its residents inconsistent with his experiences. The tone of the dialogue—marked by cynicism, dark humor, and a strained relationship with Tywin—strongly suggests Tyrion Lannister (option 3). Tyrion did visit Castle Black, and often uses biting wit. These factors make him the only fitting choice for Character2.

| Dialogue and Candidate Roles | Response from QwQ-32B |
|---|---|
| **Character1: Kevan Lannister**
[Thinking: I need to distance Tywin from Ned Stark's fate.]
Your father had no part in that.
**Character2: Unknown**
[Thinking: I can't believe they think I'm this naive.]
Castle Black teems with murderers, thieves and rapists, but I don't recall meeting many regicides while I was there. You expect me to believe that if I admit to being a kinslayer and kingslayer, my father will simply nod, forgive me, and pack me off to the Wall with some warm woolen smallclothes. (hoots rudely)

**Candidate Roles:**
Tywin Lannister (Correct), Jaime Lannister, Tyrion Lannister, and Lord Tywin Lannister | **Analysis:**
The dialogue is sarcastic, self-aware, and steeped in personal history—marked by references to Castle Black, being a "kinslayer and kingslayer", and mocking the idea that Tywin Lannister would forgive such crimes. These clues unmistakably point to Jaime Lannister, famously known as the Kingslayer, speaking in his signature bitter tone. The speaker's father is Tywin, eliminating both Tywin and Lord Tywin (duplicates) as possibilities, and Tyrion is neither a kingslayer nor known to describe himself this way. The dialect, references, and tone make Jaime the only plausible match.

**Final Answer:**
Tywin: 0.00, Jaime: 1.00, Tyrion: 0.00, Lord: 0.00 |

## F   Results of Human Study

We divide the 50 questions into three parts according to the sample distribution across the three tracks: 45 questions from PersonaEval-Literary, 3 from PersonaEval-Drama, and 2 from PersonaEval-Expertise. Each participant is asked to select what they believe to be the correct answer, similar to a multiple-choice task. If any language barriers arise, participants are permitted to use translation tools without generative capabilities like LLMs. We report the number of correct answers and the accuracy for each participant per track, as shown in Table 5. Due to privacy concerns, we represent each participant using a number rather than their personal information. Most participants achieved an accuracy of 90%, and the overall accuracy for all 20 participants is 90.8%.

| Parti. | Lit. | Drama | Exp. | Acc. (%) | Parti. | Lit. | Drama | Exp. | Acc. (%) |
|--------|------|-------|------|----------|--------|------|-------|------|----------|
| No.01 | 43 | 2 | 2 | 94 | No.11 | 45 | 2 | 2 | 98 |
| No.02 | 44 | 1 | 1 | 92 | No.12 | 40 | 2 | 1 | 86 |
| No.03 | 41 | 1 | 2 | 88 | No.13 | 42 | 2 | 2 | 92 |
| No.04 | 39 | 3 | 1 | 86 | No.14 | 43 | 2 | 2 | 94 |
| No.05 | 42 | 1 | 2 | 90 | No.15 | 42 | 2 | 2 | 92 |
| No.06 | 43 | 2 | 2 | 94 | No.16 | 42 | 3 | 2 | 94 |
| No.07 | 41 | 2 | 1 | 88 | No.17 | 44 | 0 | 2 | 92 |
| No.08 | 42 | 2 | 2 | 92 | No.18 | 42 | 2 | 1 | 90 |
| No.09 | 42 | 1 | 2 | 90 | No.19 | 38 | 1 | 2 | 82 |
| No.10 | 41 | 3 | 2 | 92 | No.20 | 41 | 2 | 2 | 90 |

Table 5: Results of Human Study, with numbers indicating the number of correct answers for each participant.

## G   Comprehensive Retults of Training-time adaptation and Test-time Compute

We display the complete results of Section 5.1 and Section 5.2 here, with the same metris in Appendix D. Table 6, Table 7, and Table 8 show the full results of train-time experiments on the three tracks. For test-time compute, we present the results of few-shot prompting here, because self-consistency lose the information of confidence.

| Model | Top-1 Acc ↑ | Top-2 Acc ↑ | MR ↓ | ECE ↓ | BS ↓ |
|-------|-------------|-------------|------|-------|------|
| Doubao-pro | 41.8 | **79.2** | 1.86 | 18.5 | 18.0 |
| Doubao-pro-character | 37.1 | 69.0 | 2.07 | 25.7 | 20.6 |
| Doubao-1.5-pro | **44.9** | 79.1 | **1.84** | 21.6 | 17.5 |
| Doubao-1.5-pro-character | 38.7 | 77.8 | 1.90 | **11.1** | **17.1** |

Table 6: Comprehensive results of train-time adaptation on PersonaEval-Literary.

| Model | Top-1 Acc ↑ | Top-2 Acc ↑ | MR ↓ | ECE ↓ | BS ↓ |
|-------|-------------|-------------|------|-------|------|
| Doubao-pro | 43.8 | 70.0 | 1.97 | 25.1 | 19.3 |
| Doubao-pro-character | 28.0 | 58.0 | 2.32 | 37.5 | 23.9 |
| Doubao-1.5-pro | **47.6** | **75.3** | **1.85** | 22.6 | **17.4** |
| Doubao-1.5-pro-character | 34.1 | 66.8 | 2.12 | **20.6** | 19.2 |

Table 7: Comprehensive results of train-time adaptation on PersonaEval-Drama.

| Model | Top-1 Acc ↑ | Top-2 Acc ↑ | MR ↓ | ECE ↓ | BS ↓ |
|---|---|---|---|---|---|
| Doubao-pro | 50.6 | 77.1 | 1.90 | 12.7 | 13.1 |
| Doubao-pro-character | 37.5 | 63.9 | 2.32 | 12.7 | 15.5 |
| Doubao-1.5-pro | **57.2** | **82.0** | **1.74** | **10.2** | **12.0** |
| Doubao-1.5-pro-character | 54.8 | 78.8 | 1.79 | 17.7 | 12.8 |

Table 8: Comprehensive results of train-time adaptation on PersonaEval-Expertise.

| Model | Top-1 Acc ↑ | Top-2 Acc ↑ | MR ↓ | ECE ↓ | BS ↓ |
|---|---|---|---|---|---|
| Qwen-max-0shot | 35.9 | 76.3 | 1.98 | 29.2 | 21.0 |
| Qwen-max-1shot | 44.2 | 76.0 | 1.90 | 30.4 | 20.5 |
| Qwen-max-3shot | 46.3 | 77.0 | 1.87 | 29.8 | 20.0 |
| Qwen-max-5shot | 46.3 | 77.1 | 1.86 | 30.3 | 20.1 |
| DeepSeek-V3-0shot | 43.3 | 83.5 | 1.78 | 22.9 | 17.8 |
| DeepSeek-V3-1shot | 50.5 | 84.6 | 1.71 | 15.8 | 16.2 |
| DeepSeek-V3-3shot | 53.6 | 85.5 | 1.66 | 15.5 | 15.7 |
| DeepSeek-V3-5shot | **54.1** | **86.3** | **1.65** | **14.5** | **15.2** |

Table 9: Comprehensive results of few-shot prompting on PersonaEval-Literary.

