# OpenReview forum: "PersonaEval: Are LLM Evaluators Human Enough to Judge Role-Play?"
_colmweb.org/COLM/2025/Conference — COLM 2025_

### Official Review · Reviewer_m7sw · 2025-05-12

**Rating:** 7
**Confidence:** 4
**Ethics Flag:** 1

**Summary:**

This paper introduces PersonaEval, the first benchmark to evaluate whether LLMs can identify human roles from natural dialogue, a necessary but underexplored foundation for reliable role-play evaluation. The authors demonstrate that even the best-performing LLMs significantly underperform compared to humans on this task. Through experiments, they show that improvements of role-play evaluation stem more from enhanced reasoning and test-time compute rather than role-specific training/prompting.
Overall, this work addresses an underexplored weakness in current LLM evaluators, offering a new benchmark, insightful empirical findings, and future directions for inducing cognitive capabilities of LLM evaluators.

**Questions To Authors:**

1. As mentioned, each participant is given 50 examples sampled from cases where DeepSeek-R1 makes incorrect high-confidence predictions. However, since the sample distribution differs between humans and LLMs, is this setting fair?

2. It is not very clear what kinds of reasoning capabilities are most helpful for role identification.

3. The Gemini-2.5-pro-0325 model is frequently used (in Figure 1 & case study). It is unclear why is omitted from the main results in Figure 3.

**Reasons To Accept:**

1. The paper tackles an underexplored problem in human-aligned role-play evaluation: the ability of LLMs to accurately identify human roles in dialogue.

2. The authors propose the first benchmark targeting role identification in natural dialogues.

3. They conduct both LLM experiments and human evaluations, revealing a substantial performance gap between state-of-the-art LLMs and humans.

4. The authors find an interesting insight that test-time reasoning capability plays a more critical role in this task than role-specific training or prompting.

**Reasons To Reject:**

1. While the paper identifies an important weakness in LLM evaluators, its focus is narrowly confined to the task of role identification. It remains unclear how performance on this task directly impacts the overall quality of role-play evaluations.

2. The paper lacks sufficient detail on training-time adaptation, and the test-time compute strategies discussed are limited.

---

> ### Author Response · Authors · 2025-06-02
>
> Thank you very much for your thoughtful review and insightful questions.
>
> **About the Scope of Role Identification and Its Impact:**
>
> Thank you for the important question. We agree it is crucial to clarify why speaker (role) identification is fundamental for role-play evaluation.
>
> In role-play evaluation, LLM judges are expected to assess whether an agent consistently adheres to its assigned role throughout a dialogue. Accurate role identification enables LLMs to correctly interpret each utterance by understanding the speaker's identity, stance, and communicative intent. This grounding is essential to avoid misattributing behaviors or inconsistencies to the wrong roles, which would otherwise compromise the fairness, precision, and interpretability of evaluations.
>
> Without reliable role identification, LLMs may conflate speakers, misjudge role fidelity, or introduce spurious errors. Therefore, while role identification is not the final goal, it is a necessary precondition — ensuring that evaluation judgments are based on an accurate understanding of dialogue structure and dynamics, thereby supporting the credibility of role-play evaluation outcomes.
>
> Our primary goal is to bring attention to this under-explored but critical prerequisite and to systematically examine current LLMs' capabilities in this regard. We sincerely appreciate your deep engagement with our work and will revise the paper to articulate this connection more explicitly.
>
> **About Training-time Adaptation:**
>
> Thank you for highlighting this important point. The models we use for the experiments are closed-source and provided by ByteDance. The training-time adaptation is performed by fine-tuning on role-play-related dialogue data. Although there is currently no official English publication describing the exact training procedure for the Doubao series, we refer to available technical documentation provided by the developers. We consult [this document (in Chinese)](https://www.volcengine.com/docs/82379/1263277) and translate relevant sections for our understanding.
>
> Based on the available information, the adaptation process involves fine-tuning on datasets enriched with role knowledge, identity recognition, dialogue alignment, behavior alignment, and instruction tuning. These are implemented under a broader System Prompt + SFT framework to ensure that models can maintain persona consistency, recognize social roles, and infer identity throughout dialogue interactions. In addition, the system also emphasizes multi-turn dialogue memory, context retention, and user portrait construction to support long-term coherence.
>
> Although we do not have access to all implementation details, such as the exact data composition or fine-tuning hyper parameters, we will incorporate a summarized description of the training approach and acknowledge these limitations in the revised manuscript.
>
> **About Test-time Compute Strategies:**
>
> Thank you for the thoughtful and valuable suggestion. Due to time and resource constraints, our current exploration of test-time compute strategies is limited. We greatly appreciate the reviewer's input, and if there are specific works or methods that you recommend for further evaluation, we would be happy to consider and incorporate them.
>
> In our study, we have already included several widely recognized and representative strategies:
> - Chain-of-Thought prompting, encouraging the model to reason step-by-step, as detailed in Appendix A;
> - Few-shot prompting, providing exemplar cases to guide predictions;
> - Self-consistency decoding, aggregating multiple reasoning paths for greater stability.
>
> These strategies represent the current mainstream approaches for enhancing LLM reasoning at test time. We believe that these explorations have sufficiently supported the objectives of this section, demonstrating both the effectiveness of mainstream test-time methods and the remaining challenges. While we acknowledge that further exploration of advanced strategies such as dynamic context expansion or tree-based reasoning could provide additional insights, a comprehensive study of such methods would go beyond the intended scope of this paper. Nevertheless, we would welcome any concrete suggestions for additional methods to evaluate, and would be happy to incorporate them where feasible.

---

> > ### Comment · Reviewer_m7sw · 2025-06-07
> >
> > Thank you to the authors for their detailed responses. Most of my concerns have been addressed, so I will increase my score.

---

> ### Author Response · Authors · 2025-06-02
>
> **About Human Evaluation Fairness:**
>
> Thank you for raising this important concern. For human evaluation, we select challenging cases — specifically, examples where DeepSeek-R1 made high-confidence errors — to ensure meaningful comparisons on non-trivial instances. While this sampling introduces distributional differences, it is actually a conservative choice favoring the models: if humans perform well on cases that are consistently difficult for models, the performance gap becomes even more compelling. We have manually verified that many cases in the benchmark are straightforward for humans. If we had sampled uniformly across the full benchmark, human accuracy would likely have been even higher.
>
> **About Reasoning Capabilities for Role Identification:**
>
> Our analysis suggests that several types of reasoning are particularly important for effective role identification:
>
> - Perspective-taking: Inferring the speaker's background, goals, and point of view based on dialogue clues.
> - Intent inference: Understanding the underlying intention behind an utterance, beyond surface-level semantics.
> - Pragmatic reasoning: Interpreting the social and contextual meanings of statements within the interaction.
>
> We will expand the discussion of these aspects in the revised version to make the connection between reasoning and role identification clearer.
>
> **About Gemini-2.5-pro-0325 Results:**
>
> Gemini-2.5-pro-0325 was released shortly before our submission deadline. Due to computational resource constraints, we were unable to include full-scale results at that time. However, we have since conducted additional evaluations with newer and more models, including Gemini-2.5-pro-0506, Qwen3-235b-a22b, and Grok-3-beta. We will include these updated results in the camera-ready version.
>
>  | Model               | Top-1 Acc ↑ | Top-2 Acc ↑ | MR ↓  | ECE ↓ | BS ↓  |
> |---------------------|-------------|-------------|------|------|------|
> | gemini-2.5-pro-0506  | 68.8        | 92.6        | 1.38 | 4.9  | 10.5 |
> | qwen3-235b-a22b      | 48.8        | 84.8        | 1.69 | 20.3 | 16.6 |
> | grok-3-beta          | 56.2        | 91.0        | 1.49 | 4.4  | 12.7 |
>
> Metrics include top-1 accuracy, top-2 accuracy, mean rank (MR) of the ground truth role in responses, Expected Calibration Error (ECE), and Brier Score (BS).
>
> Thank you again for your valuable feedback, which has greatly helped us improve the clarity and completeness of our work.

---

### Official Review · Reviewer_meJF · 2025-05-12

**Rating:** 7
**Confidence:** 4
**Ethics Flag:** 1

**Summary:**

The ability of LLMs to play various roles is fundamental to many use cases. To evaluate whether LLMs are good at playing roles, previous work has frequently used LLMs to rate the role-playing capabilities of other LLMs.  This paper investigates how well LLMs compare to humans when it comes to guessing who said what in a dialogue. It is claimed that this ability is fundamental to being able to judge the role-playing capability of another LLM. The results show that in adverserially constructed and selected cases, humans do significantly better than the best LLMs. It is argued that the reason is that guessing who said what relies on inference rather than surface cues, and LLMs tend to overuse the latter.

**Questions To Authors:**

Line 72: Role-playing *is* essential

Line 93: "To avoid the subjectivity bias originated from scoring, some works let LLM evaluators to perform...": remove "to"

Great that an example is included (Section 4.3). Would have been even better if one could also see human performance on the same task, including the reasoning behind it.

**Reasons To Accept:**

The paper is a healthy and necessary challenge to untested assumptions underlying work in the field. Unless LLMs can be proven to provide judgements of sufficient quality (which often means approximating human judgements), they cannot reliably be used to rate other LLMs (or anything else for that matter).

**Reasons To Reject:**

The authors could be clearer about the connection between the ability to guess who said what and the ability to rate role-playing capabilities. It seems plausible enough but probably an explicit argument could make this into more than a hunch, which would strengthen the paper.

---

> ### Author Response · Authors · 2025-06-01
>
> Thank you for the encouraging feedback and helpful suggestions.
>
> **About Connection Between Speaker Identification and Role-Play Evaluation:**
>
> We sincerely appreciate the request for a clearer argument. Our view is that accurate speaker identification involves understanding role-specific intentions, tone, and behavior — skills that form the foundation for higher-level evaluation of role fidelity. Without reliably inferring "who" is speaking, it would be difficult to ensure that assessments of "how well" the role is played align with human judgment. We will work to make this connection more explicit in the revision.
>
> **About Example and Human Reasoning:**
>
> Thank you for the helpful suggestion. We agree this would improve the clarity of our work. In the current case study section, we briefly mention aspects of human reasoning, but due to space constraints, we could not elaborate. We plan to extend the appendix with more detailed human reasoning processes alongside the example cases.
>
> **About Minor Wording Correction:**
>
> Thank you for pointing this out. We will carefully correct the phrasing issue.
>
> Thank you again for the valuable feedback, which has been very helpful in guiding improvements to our work.

---

> > ### Comment · Reviewer_meJF · 2025-06-09
> >
> > Thank you for the clarifying responses.

---

### Official Review · Reviewer_ECv8 · 2025-05-13

**Rating:** 6
**Confidence:** 4
**Ethics Flag:** 1

**Summary:**

The paper introduces PersonaEval, a benchmark designed to evaluate whether large language models (LLMs) can reliably identify character roles from natural dialogue. This task is crucial for human-aligned role-play evaluation, as it tests the model's ability to recognize who is speaking based on dialogue context. The benchmark uses human-authored dialogues from novels, scripts, and video transcripts, challenging models to select the correct persona from carefully constructed distractors. Experiments, including a human study, show that even the best-performing LLMs achieve only around 65% accuracy, well below the human performance of 90.8%. The authors investigate training-time adaptation and test-time compute strategies, finding that reasoning models and test-time methods show more promise than role-specific training.

**Questions To Authors:**

- Title Scope Concern: Isn't your title "LLM-as-a-judge Enough to Judge Role-Play?" overly broad compared to your actual research focus on speaker identification in role-play contexts? While speaker identification constitutes one dimension of role-play assessment, the comprehensive evaluative scope suggested by your title may create expectations among readers that exceed the actual parameters of your investigation.
- Data Filtering Impact: Could you address the substantial data reduction implemented in Stage 2 (Section 3.4) and its potential effects on your evaluation results? Specifically, have you analyzed whether this filtering procedure might disproportionately disadvantage certain models, particularly those in the Qwen series?

**Reasons To Accept:**

1. The benchmark is well-designed with a clear task formulation and a robust evaluation metric.
2. The benchmark and findings have the potential to significantly impact the field by highlighting the limitations of current LLM evaluators and guiding future research.
3. The paper is well-written and clearly structured, making it easy to follow the methodology and results.

**Reasons To Reject:**

1. Data quality and its utility for future applications are paramount considerations for any data-oriented research endeavor. The Hard Case Curation methodology, particularly in Stage 2, relies exclusively on confidence metrics from a single specific selection model, potentially compromising reliability and validity. Consequently, we can only assert with certainty that the retained samples present challenges specifically for the Qwen2.5 model series. This selection procedure lacks cross-validation protocols and may introduce systematic biases in the resulting dataset.
2. While the benchmark is well-designed, the paper could benefit from a more detailed discussion of the potential biases in the data sources and how they might affect the evaluation results.

---

> ### Author Response · Authors · 2025-06-01
>
> Thank you very much for the detailed and insightful review. We sincerely appreciate the thoughtful feedback and suggestions.
>
> **About Hard Case Curation and Data Filtering Impact:**
>
> Thank you for your valuable suggestions regarding data distribution and potential biases — we find them very important and constructive.
>
> We carefully inspect the unfiltered dataset and observe that many trivial cases exist — for instance, simple greetings or direct mentions of the speaker's name, which make the answer obvious and offer little insight into real role identification challenges. To ensure the benchmark focuses on meaningful reasoning, hard case curation is necessary.
>
> Due to the large scale of the data and limited human resources, manual filtering is impractical. Therefore, we opt for an automated and efficient filtering strategy using a relatively strong model (Qwen-max). Importantly, we do not directly select cases where Qwen fails, but rather use a confidence threshold (less than 0.5 for the correct answer) to identify uncertain cases, reducing overfitting the filter to any particular model's errors.
>
> In this work, we primarily focus on uncovering an overlooked problem in LLM evaluators. While we acknowledge that a more carefully designed filtering strategy could further improve the dataset, our current approach of selecting sufficiently challenging cases already provides strong support for the main claims we aim to make. Nonetheless, we agree that incorporating multi-model cross-validation would enhance the robustness and general utility of the benchmark for future applications, and we will further refine the filtering process to enhance data quality.
>
> **About Data Source Bias:**
>
> We appreciate your point regarding potential biases in data sources. In constructing the benchmark, we select diverse, human-authored dialogues — including novels, scripts, and video transcripts — to cover a range of domains and narrative styles commonly associated with role-play. Our preliminary inspection suggests that literary texts emphasize character personality and inner thoughts, while scripts focus more on conversational style, and instructional videos involve role-audience alignment. This diversity helps mitigate domain-specific bias to some extent. That said, we agree that a more systematic analysis of possible genre or cultural biases would further strengthen the work, and we plan to expand this discussion and analysis in the final version.
>
> **About Title Scope:**
>
> Thank you for pointing this out. Our intent is to emphasize the critical importance of role identification as a necessary condition for credible role-play evaluation. While this task represents only one aspect of role-play evaluation, it is foundational: without reliably identifying who is speaking, assessing higher-level role fidelity becomes difficult. That said, we agree that the current title could create broader expectations. Following your suggestion, we will consider revising the title to better reflect the specific focus, for example, "PersonaEval: Evaluating LLM Evaluators of Role-Play".
>
> Thank you again for your constructive feedback, which has greatly helped us refine and improve the clarity and scope of our work.

---

### Official Review · Reviewer_kR5i · 2025-05-14

**Rating:** 6
**Confidence:** 4
**Ethics Flag:** 1

**Summary:**

This paper shared a LLM Benchmark that is aiming to identify the speaker of a turn given the context and content. The benchmark contains 3 tracks, with adversarial distractor and hard case curation.

They compared LLM performance and human evaluation with accuracy. It showed human can achieve 90.8% precision while the best LLM can achieve 64.8%, with a significant gap.

The author also tested with injecting the role knowledge with SFT, or using few-shot prompting and self-consistency. The results showed SFT doesn't change much, while few-shot can improve the performance.

Overall, I like the paper.

* I agree that identifying the speaker is the first step toward better evaluate the role fidelity.
* The paper is written clearly, structured well.
* The dataset curation makes sense, with enough diversity, with adversarial data and hard case considered.
* The experiment is straightforward, the results kinds of makes sense. The SFT/Few-shot/Self-consistency results also make sense.

What made me hesitate is that:
* The task set up: wether given two turns of conversation, then identifying the speaker from a choice of 4 is a typical set up. Under what circumstances, we need to use this eval? I image, in a role play conversation, we know who is in the conversations, and the context we have is more than just 2 turns.

**Reasons To Accept:**

The + part in the summary. This task seems to be a useful one, and benchmark could be useful.

**Reasons To Reject:**

The - part in the summary.

* The task set up made me concerned about the value of the work.
* The dataset curation about language distribution, as there are 2 English track, 1 Chinese track. Is there any consideration behind it?

---

> ### Author Response · Authors · 2025-06-01
>
> Thank you for the thoughtful review and valuable feedback.
>
> **About Task Setup:**
>
> We appreciate the reviewer's concern regarding the two-turn, four-choice setup. Our design aims to focus on a fundamental prerequisite for role-play evaluation: the ability to accurately identify "who" is speaking. We believe that if an LLM cannot reliably perform role identification, it would be difficult for it to credibly assess higher-level aspects of role fidelity.
>
> While this setting may appear less common, this is largely because prior work has often implicitly assumed that LLMs possess such basic capabilities. Our intention is to carefully examine this assumption and bridge a gap that, to our knowledge, has been under-explored. We agree that extending evaluations to richer, multi-turn contexts is important, and we view our work as a necessary first step toward that broader goal.
>
> **About Language Distribution:**
>
> Thank you for raising this important question. In designing the benchmark, we aim to balance language diversity with the availability and quality of human-authored materials. The current setup — two English tracks and one Chinese track — reflects both practical and strategic considerations. English resources, especially literary novels, are more abundant and varied, allowing us to construct robust, high-quality benchmarks. At the same time, we include a Chinese track to introduce linguistic and cultural diversity, recognizing the growing importance of evaluating models in non-English contexts.
>
> Additionally, the three tracks are intentionally constructed from different types of texts — literary novels (PersonaEval-Literary), screenplays (PersonaEval-Drama), and instructional dialogues (PersonaEval-Expertise) — to cover a range of role-play related contexts beyond language diversity.
>
> We agree that expanding to more languages and domains would further strengthen the benchmark, and we plan to pursue this in future updates.
>
> Thank you again for your constructive feedback, which has been invaluable in helping us refine and clarify our contributions.

---

### Decision · Program_Chairs · 2025-07-08

**Decision:**

Accept

**Comment:**

This paper presents PersonaEval - a benchmark with 28,565 examples curated from real human data and augmented with distractor multiple choice answers to test if LLMs can identify human roles. The paper then tests a set of LLMs on PersonaEval and notes a performance gap between the best performing LLMs (65%) and humans (91%).

All reviewers identified value in this work:
* The paper proposes a potentially useful benchmark (kR5i) that offers a “healthy and necessary challenge to untested assumptions underlying work in the field” (meJF) and represents the “first benchmark targeting role identification in natural dialogues." (M7sw)
* Reasonable dataset curation approach (ECv8, kR5i)
* Insightful and potentially impactful findings (e.g., the gap between humans and existing models) (ECv8, m7sw)
* Clear presentation (kR5i, ECv8)

Overall, the strengths are that the benchmark elicits a clear gap in human/model performance and that it was collected following a reasonable methodology to augment human data (from three sources) with other multiple choice questions. Reviewers had some remaining questions about the utility of the dataset specifically its match to downstream tasks:
* “The authors could be clearer about the connection between the ability to guess who said what and the ability to rate role-playing capabilities.” (meJF)
* “It remains unclear how performance on this task directly impacts the overall quality of role-play evaluations." (m7sw)
* “Under what circumstances, we need to use this eval?” “The task set up made me concerned about the value of the work.” (kR5i)
And had other questions about specific dataset setup & results (e.g., 2 English and 1 Chinese source, approach to filtering data). Ultimately, all reviewers were at least slightly in favor of accepting this work due to its strengths.